# Intra- and Inter-Modular Connectivity Alterations in the Brain Structural Network of Spinocerebellar Ataxia Type 3

**DOI:** 10.3390/e21030317

**Published:** 2019-03-23

**Authors:** Chi-Wen Jao, Bing-Wen Soong, Tzu-Yun Wang, Hsiu-Mei Wu, Chia-Feng Lu, Po-Shan Wang, Yu-Te Wu

**Affiliations:** 1Brain Research Center, National Yang-Ming University, Taipei 112, Taiwan; 2Institute of Biophotonics, National Yang-Ming University, Taipei 112, Taiwan; 3Department of Neurology, Shuang Ho Hospital, New Taipei City 235, Taiwan; 4Institute of Neuroscience, Taipei Medical University, Taipei 110, Taiwan; 5Department of Biomedical Imaging and Radiological Sciences, National Yang-Ming University, Taipei 112, Taiwan; 6Department of Neurology, Taipei Veterans General Hospital, Taipei 112, Taiwan; 7Department of Neurology, Taipei Municipal Gan-Dau Hospital, Taipei 112, Taiwan

**Keywords:** spinocerebellar ataxia type 3, modular analysis, structural connectivity, cerebral cortices

## Abstract

In addition to cerebellar degeneration symptoms, patients with spinocerebellar ataxia type 3 (SCA3) exhibit extensive involvements with damage in the prefrontal cortex. A network model has been proposed for investigating the structural organization and functional mechanisms of clinical brain disorders. For neural degenerative diseases, a cortical feature-based structural connectivity network can locate cortical atrophied regions and indicate how their connectivity and functions may change. The brain network of SCA3 has been minimally explored. In this study, we investigated this network by enrolling 48 patients with SCA3 and 48 healthy subjects. A novel three-dimensional fractal dimension-based network was proposed to detect differences in network parameters between the groups. Copula correlations and modular analysis were then employed to categorize and construct the structural networks. Patients with SCA3 exhibited significant lateralized atrophy in the left supratentorial regions and significantly lower modularity values. Their cerebellar regions were dissociated from higher-level brain networks, and demonstrated decreased intra-modular connectivity in all lobes, but increased inter-modular connectivity in the frontal and parietal lobes. Our results suggest that the brain networks of patients with SCA3 may be reorganized in these regions, with the introduction of certain compensatory mechanisms in the cerebral cortex to minimize their cognitive impairment syndrome.

## 1. Introduction

Spinocerebellar ataxia type 3 (SCA3) is an inherited neurodegenerative disorder caused by CAG expansion in the coding region of chromosome 14q32.1 [1]. Clinically, patients with SCA3 exhibit cerebellar syndrome, including ataxia of gait, dysarthria, dysmetria, nystagmus, peripheral neuropathy, and pyramidal and extrapyramidal manifestations [2]. These patients also exhibit cognitive impairment and emotional deficits, which may be caused by not only cerebellar but also cerebral cortex atrophy [3]. The mature human brain is both structurally and functionally specialized, such that discrete areas of the cerebral cortex perform distinct types of information processing [4]. Network analyses have revealed that the organization of the brain’s structural connections enables efficient information processing and thereby supports complex brain functions [5]. By using a modular network, the relevant substructures serving specific functions can be identified, thus establishing a link between brain structure and function [6]. Furthermore, a modular network with graph theory can provide further quantitative insight into the organization of complex brain networks and their regional connections [7]. Network studies have revealed that the cerebellum has strong connections with the cerebral cortices, particularly the frontal lobe [8], and that the cortico-cerebellar system plays a crucial role in motor learning and relearning [9].

A cortical feature-based network can manifest the anatomical connections between parcellated regions in neural degenerative diseases such as dementia [10], multiple sclerosis [11], and Alzheimer disease [12]. Cortical morphology features, including thickness [13], volume [14], and surface area [15], have been applied in constructing structural connectivity networks. However, the structural networks of patients with SCA3 and their connecting patterns remain unclear and minimally explored. Fractal dimension (FD) analysis has been extensively used to quantify shape complexity and morphological changes in cerebral magnetic resonance imaging (MRI) [16]. An FD descriptor can offer quantitative information related to cortical convolution, and changes in the FD value reportedly indicates cortical abnormalities [16]. Thus, an FD value may serve as a suitable feature in developing a structural network.

However, the brain structural network of patients with SCA3 has been minimally explored. In this study, we aimed to investigate structural connectivity in patients with SCA3. We proposed a novel 3D-FD-based structural network for healthy and SCA3 study participants. The automated anatomical labeling (AAL) atlas was used to parcellate the cerebrum and cerebellum into 97 subregions as relatively focal regions. First, FD values for the 97 parcellated regions in the control and SCA3 groups were measured. We then calculated the copular correlation of 3D-FD values between paired brain regions to produce a 97 × 97 correlation matrix for both groups. These correlation matrices were subsequently used in modular analysis to establish a structural connectivity network and compute the intra-modular and inter-modular connectivity of the whole brain and its lobes. We hypothesized that alterations of such connectivity could reflect cortical involvement in both non-motor functions and motor networks in patients with SCA3.

## 2. Materials and Methods

### 2.1. Participants

A total of 48 patients with SCA3 and 48 healthy controls participated in this study. All of the participants were recruited from the Department of Neurology at Taipei Veterans General Hospital from 2005 to 2012. Table 1 summarizes the demographic data of the two groups. No significant differences in age (p = 0.607) and sex were observed between the two groups. The Scale for the Assessment and Rating of Ataxia scores (SARA) for the patients with SCA3 revealed that they had walker gait status, and their Mini-Mental State Examination (MMSE) scores indicated a normal mental state. T1- and T2-weighted images of each participant were examined by an experienced neuroradiologist. Cerebral atrophy was observed in nine patients with SCA3, whereas 39 patients exhibited cerebellar atrophy. All control group participants had no diseases of the central nervous system and did not exhibit any neurological abnormalities during the study period. The study protocol was approved by the Institutional Review Board of Taipei Veterans General Hospital, and written informed consent was obtained from each participant.

### 2.2. Image Acquisition and Cortical Feature-Based Structural Network 

The brain axial MRI encompassing the entire cerebrum and cerebellum was performed using a 1.5-T Vision scanner (Siemens, Erlangen, Germany). The participants were scanned using a circularly polarized head coil to obtain T1-weighted images (repetition time, 14.4 ms; echo time, 5.5 ms; matrix size: 256 × 256; 1.5-mm axial slices; field of view, 256 × 256 mm; voxel size, 1.0 × 1.0 × 1.5 mm^3^). Each structural MRI dataset was normalized to a pre-segmented and validated volumetric template by using DiffeoMap, and each normalized image volume was then segmented into gray matter, white matter, and cerebral spinal fluid in the native space by using the SPM5 toolbox. Each voxel of gray matter was then anatomically aligned to the 116 Automated Anatomical Labeling (AAL) structures by using IBASPM (Individual Brain Atlases using the Statistical Parametric Mapping) toolbox in MATLAB R2013b software (Mathworks, Natick, MA, USA). In this study, the 26 regions of the cerebellum were merged to seven regions according to their anatomical structures. Thus, parcellation divided the cortical cortex into 97 regions (Appendix A).

### 2.3. FD Analysis and Brain Structural Network

FD is a quantitative indicator of object complexity. Many studies have used FD to investigate morphological changes in the cerebral cortex caused by neurological diseases [16,17,18]. The procedure of computing FD is as follows:

Let *N*(*r*) denotes the minimal number of cubes of size r covering the fractal object. The power law relationship that defines the FD of a fractal is given by $N\left(r\right)\alpha \;r^{-FD}$ [16,17,18]. It follows that a larger *N*(*r*) or FD should cover a more irregular fractal object. We can rewrite the abovementioned equation in the form of a line, that is, log(N(r))=FD∗log1r+k, meaning the value of FD can be estimated from the slope of the line. There are three steps in the FD estimation procedure. In the first step, choose cubic boxes of a size r (edge length in pixel size) and stack them side by side to encompass the whole 3D fractal object. The count was set to one whenever a box encompasses any pixel belonging to the fractal object, and zero otherwise. In the second step, calculate the total number of non-empty boxes *N*(*r*) required to completely cover the whole fractal object. Progressively decrease the size r of the boxes and repeat the same counting process. In the final step, the estimated FD value is the slope value of the regression line between the log (*N*(*r*)) and log (1/*r*). Our previous studies have demonstrated that the degree of cortical degeneration, which was quantified using FD analysis, can indicate the severity of neurodegenerative disorders [16]. Because FD analysis is based on a logarithmic scale, a small increase in the FD value corresponds to large increase in complexity. A higher FD value indicates a more complex cerebral structure, while a decrease in the cerebral FD value may indicate a degeneration of the cerebral structure [16]. Hence, this method can be used for effectively measuring structural cortical changes in both the cerebellar and extracerebellar regions and facilitated, evaluating the severity of cortical atrophy in this SCA3 study.

There are two steps to perform for establishing the brain structural network: the copula correlation between paired brain regions was calculated to indicate the strength of interregional connections. Because the copula correlation has the advantage of preventing bias associated with asymmetric distributions [19], we used it to measure interregional connectivity between nodes. Accordingly, the brain structural network was derived from a 97 × 97 copula correlation matrix of 3D-FD values for paired regions. Secondly, we used modular analysis to categorize the different brain regions into several modules on the basis of their interregional connections [20].

### 2.4. Modular Analysis

In modular analysis, brain regions are categorized into several modules, where the connections are stronger within each module and weaker between modules [20]. We defined the 97 anatomical brain regions as nodes and interregional copula correlation coefficients as the edges between the nodes. The suitability of a modular partition can be measured using modularity, *Q*:
(1)$$Q=\frac{1}{2m}\sum_{i,j}\left[A_{i,j}-\frac{k_{i}k_{j}}{2m}\right]\delta \left(c_{i},c_{j}\right)$$where *A* is the connection matrix of the network, and each element of A is the copula correlation coefficient between regions; $k_{i}=\sum_{j}A_{ij}$ is defined as the sum of the copula correlation coefficient between node *i* and its connected regions, and is also called the degree of node *I*; $m=\frac{1}{2}\sum_{i,j}A_{ij}$ represents the total number of edges; and *c_i_* denotes the module of node *i*. The *δ*-function *δ*(*i*, *j*) is 1 when nodes *i* and *j* belong to the same module and 0 otherwise. *Q* represents the edge numbers of all paired nodes belonging to the same module. A larger *Q* implies a superior partition that is more likely to construct a modular organization [21].

In this study, we have set a proportional value of 0.2 as a threshold to filter the connectivity matrix by preserving 20% proportion of the strongest correlation coefficients. In this process, all other entries below the threshold, negative correlations and all entries on the main diagonal (self-to-self connections) are set to 0 and the links will not exist. The participation coefficient and intra-modular degree for each region were defined as the indices of their inter- and intra-modular connection density, respectively [22].

#### 2.4.1. Intra-Modular Connectivity Analysis

Intra-modular connectivity is measured using the normalized within-module degree:
(2)$$z_{i}=\frac{k_{i}-\bar{k_{ci}}}{\sigma_{k_{ci}}}$$where $k_{i}$ is the number of edges linking the *i*th node to other nodes in its module c; $\bar{k_{ci}}$ is the average of $k_{i}$ of all nodes in module c; and $\sigma_{k_{ci}}$ is the standard deviation of the intra-modular degrees of all nodes in module c. Thus, a higher value *z_i_* represents a stronger intra-modular connectivity for node *i*. In addition, nodes with high intra-modular degrees are considered hubs, whereas the other nodes are considered non-hubs. The intra-modular connectivity for the whole network (*Z_total_*) is the mean *Z_i_* of the whole 97 nodes. Similarly, the intra-modular connectivity for each lobe, namely *Z_frontal_*, *Z_parietal_*, *Z_temporal_*, *Z_occipital_*, or *Z_cerebellum_*, represents the average of *Z_i_* within the respective lobe.

#### 2.4.2. Inter-Modular Connectivity Analysis

Inter-modular connectivity is measured using the participation coefficient,
(3)$$P_{i}=1-\sum_{c=1}^{C}{\left(\frac{k_{ci}}{k_{i}}\right)}^{2}$$where $k_{ci}$ is the number of edges connecting the *i*th node to other nodes in its module *c*, and $k_{i}$ is the number of degrees in node *i* in the network. The participation coefficient ranges from 0 to 1. A participation coefficient of 0 represents a largely intra-modular connection within node *i*, whereas that close to 1 indicates a largely inter-modular connection of node *i*. The nodes can further be classified into connectors and provincial nodes on the basis of high and low participation coefficients, respectively. The inter-modular connectivity for the whole network (*P_total_*) is the mean *P_i_* of the whole 97 nodes. Similarly, the inter-modular connectivity for each lobe, namely *P_frontal_*, *P_parietal_*, *P_temporal_*, *P_occipital_*, or *P_cerebellum_*, represents the average of *P_i_* within the respective lobe.

### 2.5. Statistical Analysis

In this study, a two-tailed *t*-test was used to determine whether significant differences existed between the control and SCA3 groups for 3D-FD values and network parameters, including modularity and intra- and inter-modular connectivity. Note that we only obtained one 3D-FD value for each parcellated region. For each group with 48 subjects, there are 48 3D-FD values for each region. We computed the 3D-FD value based copular correlation between any two regions. As a result, a 97 by 97 correlation map was obtained for each group to build a structural network, resulting in one set of topological properties for each structural network. Accordingly, we cannot directly perform any statistical comparison on the corresponding topological properties between these two structural networks. To statistically compare the differences of network properties between the SCA3 and control groups, a permutation test was conducted [23]. In the process, we set the *P* value range from 0.05 to 0.001 and the network properties at each *P* value were computed for the SCA3 and control groups. To test the null hypothesis that network property differences between the groups occurred by chance, we randomly reassigned the SCA3 patients and healthy controls into two groups and recomputed the correlation matrix for each randomised group. This randomised simulation and recalculation of the network properties was repeated 1000 times. The 95th percentile points of each distribution of the 1000 simulations were used as critical values in a two-sample one-tailed *t*-test to reject the null hypothesis with a type I error probability of 0.05. Then, the network properties Q, P, and Z were calculated for each reassigned correlation matrix of the two groups. Following the permutation process, we obtained the modularity (Q) value and the intra- and inter-modular connectivity for each lobe within the cerebrum. This randomized simulation and recalculation of network properties was repeated 1000 times, and these 1000 sets of network parameters were used in a two-sample one-tailed *t*-test to assess significant differences between the SCA3 and control groups.

## 3. Results

### 3.1. Patients with SCA3 Exhibited Significant Lateralized Atrophy with Extensive Involvement in the Left Supratentorial Regions

Table 2 summarizes the 37 parcellated brain regions (cerebrum/cerebellum: 32/5) that were significantly atrophied in patients with SCA3 (*p* < 0.01). These significantly atrophied regions included the frontal, parietal, temporal, and occipital lobes, the limbic system, and subcortical regions. Patients with SCA3 exhibited significant lateralized atrophy in the left supratentorial regions (left/right: 26/11), particularly the occipital and temporal lobes that were all in the left hemisphere.

### 3.2. Patients with SCA3 Exhibited Lower Modularity Values and Less Dense Modular Networks

Figure 1 illustrates the copular correlation map of AAL between different lobes for normal (Figure 1a) and SCA3 groups (Figure 1b). The color bar indicates the copula correlation intensity form 0(blue) to 1(red).

Clearly, patients with SCA3 have a less dense and sparse copula correlation map. As compared the button part of Figure 1b with Figure 1a, the cerebellum (CB) of SCA3 reveals much sparser links with other lobes, and may imply their CB are functional dissociated from other lobes. The network modularity (Q) value for the control group was 0.1957, whereas the SCA3 group had a significantly lower modularity value of 0.1568 (*p* < 0.01), suggesting that its networks were comparatively less dense and efficient.

Figure 2a–e illustrates the brain modular networks of the control group, and Figure 2f–k presents the brain modular networks of the SCA3 group. In each figure, the box diagram represents the hub node, which has dense connections with other nodes, and the ball diagram represents the non-hub node, which has fewer such connections.

The three major functional modules in the brain networks for the controls were labeled Modules I–III. Module I (Figure 2a) consisted of 48 brain regions, including the entire cerebellar cortex, thalamus (THA), caudate nucleus (CAU), putamen (PUT), pallidum (PAL), dorsolateral prefrontal cortex, medial superior frontal gyrus, superior-medial orbitofrontal cortex, middle orbitofrontal cortex, and inferior orbitofrontal cortex. In Figure 2e, the connective paths (blue arrow, dashed and solid lines) represent regions associated with cognitive and executive functions. Module II (Figure 2b) comprises 39 regions, including the parietal, occipital, and frontal lobes, which are mainly related to sensorimotor, visual, and spatial functions. Module III (Figure 2c) comprises 10 regions, including the temporal and occipital lobes and the hippocampus (HIP), which may be related to visual object recognition [24].

Patients with SCA3 exhibited network patterns different from those of the control group. Their brain connecting links were less dense and could be grouped into four network modules. Module I (Figure 2f) included the bilateral CAU, anterior cingulate gyrus (ACC), posterior cingulate gyrus, and most of the frontal regions, which are mainly associated with cognitive and executive functions. Module II (Figure 2g) comprised 39 brain regions mainly relating to sensorimotor, visual, and spatial functions, such as the parietal cortex, including the precuneus, superior parietal gyrus, and post-central gyrus, and the occipital, frontal, and temporal lobes. Module III (Figure 2h) comprised 19 regions, including the HIP, parahippocampus (PHIP), ACC, amygdala (AMY), inferior temporal gyrus (ITG), and olfactory cortex (OLF), which may be related to mnemonic and emotional functions of the brain. Module IV encompassed the entire cerebellar region, basal ganglia, including the bilateral putamen and pallidum, and bilateral thalamus (Figure 2i).

The SCA3 group exhibited different cortico-cerebellar circuit modules (Figure 2f–k). We found that brain regions included in Module I of the control group’s brain networks (cortico-cerebellar circuit) were grouped into two modules, namely Modules I and IV, in the SCA3 group. Therefore, as illustrated in Figure 2j, we combined the two modules (Modules I and IV of SCA3) to form the cortico-cerebellar circuit network of the SCA3 group for comparison with that of the control group. The red parts in Figure 2j represent cognitive and executive functions, and the purple parts represent the cerebellum–basal ganglia circuit. Compared with the results shown in Figure 2e, the cerebellum of the SCA3 group had fewer links with other lobes (red circle) and no circuit path to the prefrontal lobe (blue dashed line in Figure 2e). Notably, the entire cerebellar region in the SCA3 group was linked only to the basal ganglia and was dissociated from the prefrontal lobe. These dissociations may exclude the cerebellum of the SCA3 group from brain networks for cognitive and executive functions. Figure 2k illustrates the whole brain networks of patients with SCA3.

### 3.3. Significantly Decreased Intra-Modular Connectivity in Patients with SCA3

Figure 3 presents intra-modular connectivity data for the control and SCA3 groups resulting from the permutation test. For the whole cerebrum network, the SCA3 group exhibited significantly lower intra-modular connectivity than the control group (*p* < 0.01). We further analyzed the intra-modular connectivity of each lobe for control and SCA3 groups. The cerebellum in SCA3 exhibited the greatest decreased intra-modular connectivity (SCA3/control = 65%, *p* < 0.001), and the occipital lobe revealed the second greatest decrease (SCA3/control = 70%, *p* < 0.001) in intra-modular connectivity. The frontal and parietal lobes in SCA3 also showed decreased intra-modular connectivity (frontal: 83%, *p* < 0.05; parietal: 88%, *p* < 0.05).

### 3.4. Significantly Increased Inter-Modular Connectivity in Patients with SCA3

In contrast to the intra-modular connectivity results, the SCA3 group exhibited increased inter-modular connectivity in the network of the whole cerebrum. Figure 4 presents the inter-modular connectivity ratios for each group from the permutation test. The SCA3 group showed an overall increase in inter-modular connectivity (SCA3/control = 108%, *p* < 0.05). The parietal lobe had the greatest increase (SCA3/control = 119%, *p* < 0.001) and the frontal lobe had the second greatest increase (SCA3/control = 116%, *p* < 0.01).

Although the SCA3 group demonstrated increased inter-modular connectivity for the whole brain, the cerebellum exhibited decreased inter-modular connectivity (SCA3/control = 80%, *p* < 0.001). On comparing whole brain networks between the control and SCA3 groups, we found that the hubs in Module I of the SCA3 group overlapped more with Module II (highlighted by blue circles in Figure 2d,k). This suggests that patients with SCA3 may have increased inter-modular connectivity in their supratentorial networks.

## 4. Discussion

Following a review of the relevant literature, we prospectively proposed a novel FD value-based structural network to investigate the alteration of brain networks of control and SCA3 participants. The main findings of this study were as follows: (i) patients with SCA3 exhibited significant lateralized atrophy with extensive involvement in the left supratentorial regions; (ii) connection modules in the SCA3 group differed from those in the control group; (iii) the SCA3 group exhibited decreased cerebral-cerebellar connectivity; and (iv) the SCA3 group exhibited reduced intra-modular connectivity in every lobe but increased inter-modular connectivity in the supratentorial networks.

### 4.1. Patients with SCA3 Exhibited Cerebellar Cognitive Affective Syndrome (CCAS) and Mnemonic and Emotional Function Preservation

Human brain networks can be arranged into several modules according to cortical thickness [13], and these modules are compatible with the functional networks resulting from functional MRI (fMRI) studies [25]. Our results were consistent with other brain structure module studies using cortical thickness [13] or human brain functional networks through fMRI [25]. We found that Module I of the control group mainly represented the path along the cortico-cerebellar [24] and cortico-basal ganglia-thalamo-cortical circuits [26], which may be associated with cognitive and executive functions. Our results indicated that the cerebellum of the control group was densely connected with the prefrontal lobe and played a major role in cognitive and executive functions. The cortico-ponto-cerebellar tract has been thoroughly explored using diffusion tensor imaging [27] and neuroanatomical tracing [25]. This tract is crucial for determining the most affected areas in patients with ataxia through evaluation of white matter integrity. In our study, Module I of the control group demonstrated associated communication between the cerebrum and cerebellum via the cortico-cerebellar [25] and cortico-basal ganglia- thalamo-cortical circuits [26].

Neuroimaging studies have reported that the cerebellum is the region most affected in patients with SCA3 [28], and cerebellar damage may result in movement-related dysfunction. Patients with SCA3 are considered to have CCAS and exhibit a deficit pattern for executive function, visual-spatial cognition, linguistic performance, and changes in emotion and personality [29]. Our results indicated a disruption of cortico-cerebral circuits in patients with SCA3, and the results may reveal an association with the CCAS hypothesis, which relates to neural circuits linking the prefrontal lobe with the cerebellum. Exacerbation of CCAS may thus contribute to cognitive deficits in patients with SCA3.

### 4.2. Increased Inter-Modular Connectivity in Supratentorial Regions in Patients with SCA3

In this study, the intra- and inter-modular connectivity patterns of patients with SCA3 revealed disease-related alteration. First, several regions of the sensorimotor, visual, and spatial modules had significantly reduced cortical complexity and intra-modular connectivity in SCA3. These occipitoparietal and associated regions are responsible for spatial and image processing [30], and damage to these regions could result in visuospatial deficits [31]. A positron emission tomography study demonstrated that occipitoparietal regions in patients with SCA3 had significantly decreased regional metabolism [32]. Alterations in the intra-modular connectivity of the sensorimotor, visual, and spatial modules in SCA3 may thus indicate declines in the corresponding coordinated functions.

Second, patients with SCA3 exhibited increased inter-modular connectivity (participation coefficient) in brain regions mainly located in the entire cerebrum; cortico-cerebellar circuits; and sensorimotor, visual, and spatial modules. Notably, regions with increased inter-modular connectivity were compatible with the involved areas of supratentorial regions in patients with SCA3. Reduced connectivity between the cerebellar cortex and cognitive-related functional modules in SCA3 could indicate a disconnection between the cerebral and cerebellar cortices. The reorganization and increased inter-modular connectivity of the supratentorial networks may have resulted from the plasticity of the brain in attempting to compensate for function loss in the cerebellum. Moreover, our previous study evidenced that the SCA3 networks were globally more vulnerable to targeted attacks than the normal controls networks because of the effects of pathological topological organization [33]. The SCA3 analysis revealed a sparser and disrupted structural network with decreased values in the largest component size, mean degree, mean density, clustering coefficient and global efficiency and increased value in characteristic path length [33]. While in the present study, we used modular analysis of network, and focused local effect between different lobes. SCA3 revealed decreased intra-modular connectivity within each lobe but increased inter-modular connectivity in frontal and parietal lobes.

However, the main limitations in this study are the lack of specific cognitive evaluations and clinical parameters for assessing CCAS in SCA. Functional MRI will be considered for further investigation of the correlation between the present results of structural network and CCAS in a future study.

## 5. Conclusions

Patients with SCA3 exhibited significant lateralized atrophy in the left supratentorial regions and significantly lower modularity values. Their cerebellar regions were dissociated from higher-level brain networks. These patients also demonstrated decreased intra-modular connectivity in all lobes, but increased inter-modular connectivity in the frontal and parietal lobes of the supratentorial regions. Our results suggest that the brain networks of patients with SCA3 may be reorganized in these regions, with the introduction of certain compensatory mechanisms in the cerebral cortex to minimize their cognitive impairment syndrome.

## Figures and Tables

**Figure 1 entropy-21-00317-f001:**
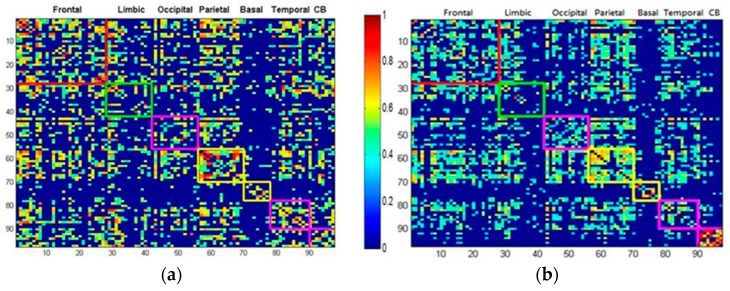
The copula correlation map of automated anatomical labeling (AAL) between different lobes. (**a**) Normal group; (**b**) SCA3.

**Figure 2 entropy-21-00317-f002:**
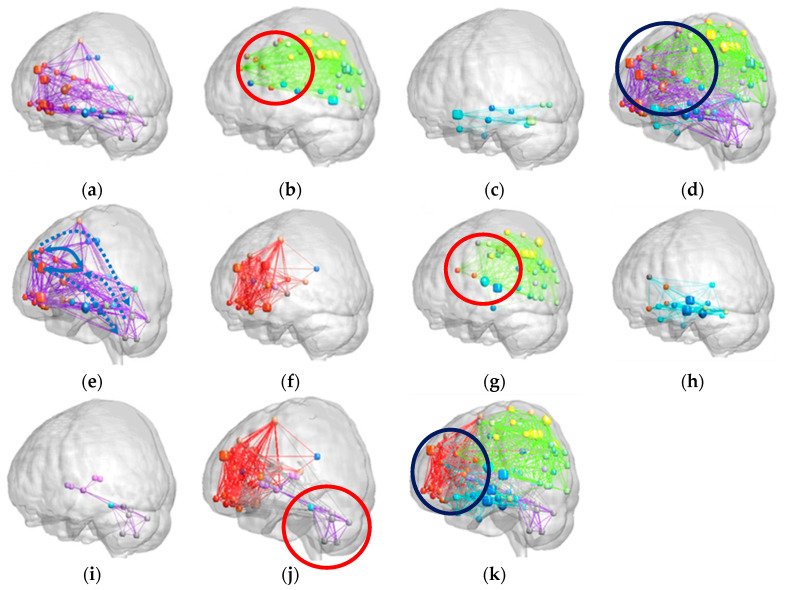
The brain networks of different modules for the controls and patients with SCA3. (**a**) Brain network module I of normal and mostly presents the path along the cortico-cerebellar circuits and cortico-basal ganglia-thalamo-cortical circuits. (**b**) Brain network module II of normal comprises the parietal, occipital, and frontal lobes. (**c**) Brain network module III of normal comprises the temporal and occipital lobes and hippocampus. (**d**) Whole brain networks of normal. (**e**) The connective paths in module I and the connections within the cortico-cerebellar circuits confirm indirect communication between the cerebral cortex and cerebellum. (**f**) Brain network module I of SCA3 patients is associated with cognitive and executive functions. (**g**) Brain network module II of SCA3 patients comprises the parietal, occipital, frontal and temporal lobes. (**h**) Brain network module III of SCA3 patients comprises the hippocampus, parahippocampus, anterior cingulate gyrus, amygdala, inferior temporal gyrus, and olfactory cortex. (**i**) Brain network module IV encompasses the entire cerebellar region; basal ganglia, including the bilateral PUT and PAL; and bilateral THA. (**j**) The combination of Module I and Module IV networks to form the cortico-cerebellar network of SCA3 patients. (**k**) Whole brain networks of SCA3 patients.

**Figure 3 entropy-21-00317-f003:**
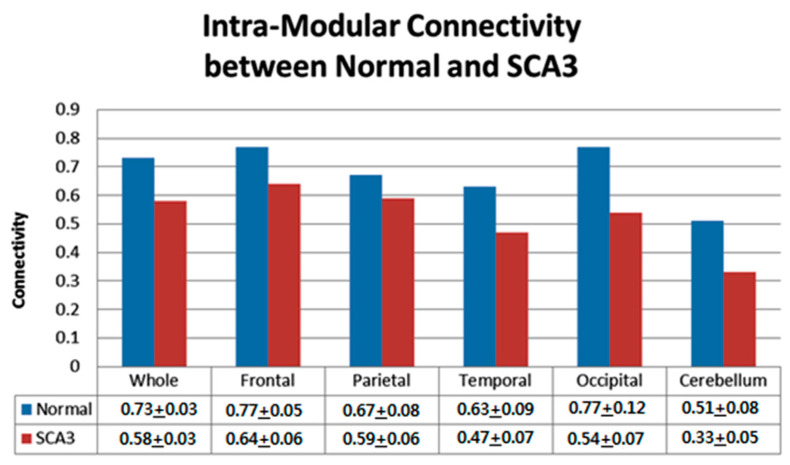
Comparison of network intra-modular connectivity for whole and each lobe between normal and SCA3 resulted from the permutation test.

**Figure 4 entropy-21-00317-f004:**
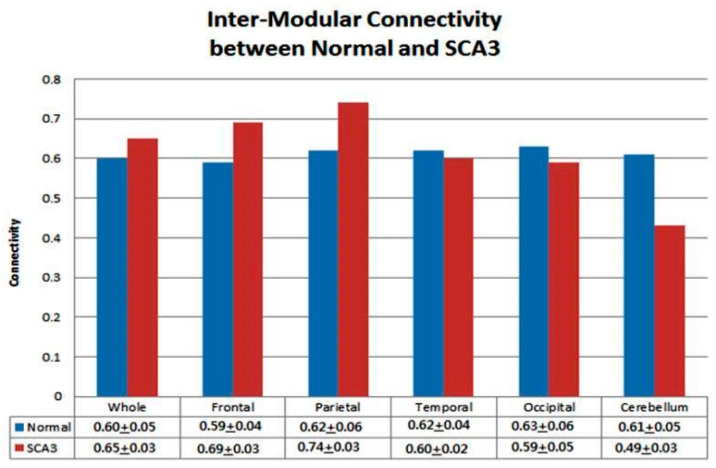
Comparison of network inter-modular connectivity for whole brain and each lobe between normal and SCA3 resulted from the permutation test.

**Table 1 entropy-21-00317-t001:** The demographic, clinical, and MR image data of control and patients with spinocerebellar ataxia type 3 (SCA3) groups.

Characteristic	Group				P
	Controls (N = 48)		SCA3 Patients (N = 48)		
	Mean	SD	Mean	SD	
**Age (years)**	48.14	12.06	48.13	11.75	0.607 ^a^
**Duration (years)**	--	--	8.89	6.43	--
**Clinical features**					
**SARA**	--	--	14	8.10	--
**MMSE**	--	--	28.5	1.61	--
**Cerebral atrophy/cerebellar atrophy** **(observed via visual inspection)**	--	--	9/39	--	--
**Gender**					
**Female**	N = 24	50%	N = 21	43.75%	0.544 ^a^
**Male**	N = 24	50%	N = 27	56.25%	0.544 ^a^

There were no significant differences in age (*p* = 0.607) and sex composition (*p* = 0.544) between SCA3 patients and controls. ^a^ two-tailed two-sample *t*-test; SARA = Scale for the Assessment and Rating of Ataxia; MMSE = Mini-Mental State Examination.

**Table 2 entropy-21-00317-t002:** Significantly atrophied brain regions in patients with SCA.

Region(L/R)	Controls	SCA3	Region (L/R)	Controls	SCA3
**Cerebellar Cortex**			**Parietal Lobe**		
Entire	2.56 ± 0.02	2.53 ± 0.04	Post-central gyrus(L)	2.17 ± 0.05	2.10 ± 0.06
Anterior lobe (L)	2.17 ± 0.04	2.11 ± 0.07	Superior parietal gyrus(L)	2.10 ± 0.06	2.03 ± 0.06
Anterior lobe (R)	2.15 ± 0.04	2.03 ± 0.08	Superior parietal gyrus(R)	2.08 ± 0.06	2.04 ± 0.07
Posterior lobe(L)	2.47 ± 0.03	2.45 ± 0.04	Inferior parietal gyrus (L)	2.19 ± 0.07	2.09 ± 0.08
Posterior lobe(R)	2.48 ± 0.03	2.44 ± 0.04	Supramarginal gyrus (L)	2.11 ± 0.05	2.04 ± 0.06
Vermis	2.15 ± 0.05	2.12 ± 0.04	Angular gyrus (L)	2.12 ± 0.07	2.00 ± 0.09
**Frontal Lobe**			Precuneus (L)	2.21 ± 0.03	2.17 ± 0.05
Precentral gyrus(L)	2.15 ± 0.07	2.07 ± 0.07	Precuneus (R)	2.14 ± 0.04	2.10 ± 0.04
Superior frontal gyrus (L)	2.08 ± 0.03	2.05 ± 0.05	**Occipital Lobe**		
Superior frontal gyrus (R)	2.13 ± 0.04	2.10 ± 0.06	Calcarine fissure and surrounding cortex (L)	2.25 ± 0.04	2.22 ± 0.04
Middle frontal gyrus (L)	2.28 ± 0.04	2.25 ± 0.04	Cuneus (L)	2.13 ± 0.04	2.11 ± 0.04
Orbitofrontal cortex(superior-medial) (L)	2.11 ± 0.04	2.08 ± 0.05	Lingual gyrus (L)	2.20 ± 0.04	2.17 ± 0.05
Orbitofrontal cortex(superior-medial) (R)	2.14 ± 0.04	2.10 ± 0.05	Superior occipital gyrus (L)	1.95 ± 0.06	1.89 ± 0.08
Inferior frontal gyrus (opercular) (R)	2.10 ± 0.05	2.07 ± 0.05	Middle occipital gyrus (L)	2.19 ± 0.05	2.12 ± 0.07
Inferior frontal gyrus (triangular) (L)	2.27 ± 0.04	2.23 ± 0.05	**Temporal lobe**		
Supplementary motor area (L)	2.19 ± 0.05	2.14 ± 0.05	Superior temporal gyrus (L)	2.18 ± 0.05	2.12 ± 0.05
Superior frontal gyrus (medial) (L)	2.17 ± 0.04	2.12 ± 0.06	Middle temporal gyrus (L)	2.34 ± 0.03	2.30 ± 0.05
Superior frontal gyrus (medial) (R)	2.09 ± 0.05	2.06 ± 0.07	**Limbic**		
Paracentral lobule (L)	2.04 ± 0.07	1.99 ± 0.08	Posterior cingulate gyrus (L)	1.99 ± 0.05	1.95 ± 0.05
Paracentral lobule (R)	1.98 ± 0.07	1.92 ± 0.08	Parahippocampal gyrus (R)	2.16 ± 0.03	2.14 ± 0.03
**Subcortical Regions**			**Subcortical Regions**		
Amygdala (R)	1.94 ± 0.04	1.97 ± 0.04	Caudate nucleus (R)	2.09 ± 0.04	2.04 ± 0.05
Caudate nucleus (L)	2.08 ± 0.05	2.05 ± 0.05	Lenticular nucleus, putamen (L)	2.11 ± 0.08	2.07 ± 0.05

Brain regions with significant difference (*p* < 0.01) in 3D-FD values between control and patients with SCA3. Significant difference under a corrected threshold of FDR = 0.05, the 3D-FD values are expressed as mean ± standard deviation; L: left, R: right.

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
