# Peer review of "Intra- and Inter-Modular Connectivity Alterations in the Brain Structural Network of Spinocerebellar Ataxia Type 3"

_entropy, 2019, doi:10.3390/e21030317_

Reviewer 1 Report

The article is globally well written, just check minor typing issues (e.g. line 356 "othalamo"). Conclusions are supported by the methodology and the results , although the core part of the discussion in merely speculative and not supported by specific cognitive evalutations and clinical parameters (evidences of CCAS in SCA) other than MMSE and SARA.

Authors should revise the discussion, stating the main limitations of the study (i.e. no useful clinical data), for instance they cannot discuss on CCAS without stating this is merely speculative (although interesting). Then they should declare the relationship between the present study and the one already and recently published (https://doi.org/10.3389/fnins.2018.00935) (which is not even cited, although the population and methodology is pretty much the same).

Author Response

Point 1: The article is globally well written, just check minor typing issues (e.g. line 356 "othalamo").

Author’s response:

We thank reviewer’s comment. We have proof read the article and corrected the typos in line "othalamo" as follows:

Module I of the control group demonstrated associated communication between the cerebrum and cerebellum via cortico-cerebellar [25] and cortico-basal ganglia-thalamo-cortical circuits [27].

Point 2: Conclusions are supported by the methodology and the results, although the core part of the discussion in merely speculative and not supported by specific cognitive evaluations and clinical parameters (evidences of CCAS in SCA) other than MMSE and SARA. Authors should revise the discussion, stating the main limitations of the study (i.e. no useful clinical data), for instance they cannot discuss on CCAS without stating this is merely speculative (although interesting).

Authors’ response:

We appreciate reviewer’s valuable suggestion. In the discussion section of revised manuscript, we emphasize the limitation of the present study, that is, we need more clinical data of function (functional MRI) to assess the association between CCAS syndrome and structural network in patients with SCA3. The revised manuscript is as follow:

1) Our results indicated a disruption of cortico-cerebral circuits in patients with SCA3, and the results may reveal an association with the CCAS hypothesis, which relates to neural circuits linking the prefrontal lobe with the cerebellum.

2) However, the main limitations in this study are the lack of specific cognitive evaluations and clinical parameters for assessing CCAS in SCA. Functional MRI will be considered for further investigating the correlation between the present results of structural network and CCAS in future study.

Point 3: Then they should declare the relationship between the present study and the one already and recently published (https://doi.org/10.3389/fnins.2018.00935) (which is not even cited, although the population and methodology is pretty much the same).

Author’s response:

We thank reviewer’s valuable suggestion. We have cited our previous study, “Impaired Efficiency and Resilience of Structural Network in Spinocerebellar Ataxia Type 3” of Front. Neurosci., 17 December 2018, as reference 34 in the revised manuscript. In this previous study, we used target attack to assess the resilience and investigate the small-world features of of SCA3 network. Results showed the SCA3 networks were more vulnerable to targeted attacks than the normal controls networks because of the effects of pathological topological organization. The SCA3 networks had significantly smaller clustering coefficients (P < 0.05) and global efficiency (P < 0.05) but larger characteristic path length (P < 0.05) than the normal controls networks, implying loss of small-world features.

While in the present study, we used modular analysis of network and focused the intra-connectivity within each lobe and inter-connectivity between lobes of SCA3. We have added a paragraph to clarify the difference between the present study and previous study in the discussion section of revised manuscript. The added paragraph is as follows:

Moreover, our previous study evidenced that the SCA3 networks were globally more vulnerable to targeted attacks than the normal controls networks because of the effects of pathological topological organisation [34]. The SCA3 revealed a more sparsity and disrupted structural network with decreased values in the largest component size, mean degree, mean density, clustering coefficient and global efficiency and increased value in characteristic path length [34]. While in the present study, we used modular analysis of network, and focused local effect between different lobes. SCA3 revealed decreased intra-modular connectivity within each lobe but increased inter-modular connectivity in frontal and parietal lobes.

34.   Wu YT, Huang SR, Jao CW, Soong BW, Lirng JF, Wu HM, Wang PS. Impaired Efficiency and Resilience of Structural Network in Spinocerebellar Ataxia Type 3. Front. Neurosci. 2018, 12,935. doi: 10.3389/fnins.2018.00935

Reviewer 2 Report

Reviewer’s comments

This is a well thought-out study into to a clinical area with relatively little previous investigation using 3D_FD to detect differences in network parameters between groups with copula correlations and modular analysis studying in the rare disease, SCA type3.

The figures and the statistical methods are good.

The results and discussion have been presented well and findings summarized well in the conclusions.

Please consider this minor revision:

-The author described FD analysis as a quantitative indicator of object complexity which can be used to investigate morphological changes in the cerebral cortex and the one 3D-FD value for each parcellated region of the brain was obtained.

              -The result (table 2) showed significantly atrophied brain regions in the patients with SCA type 3. >> To give the study face validity, please clearly describe or provide literature evidence that how the 3D-FD values represent degree of atrophy of the cerebral cortex.

Author Response

Point 1: This is a well thought-out study into to a clinical area with relatively little previous investigation using 3D_FD to detect differences in network parameters between groups with copula correlations and modular analysis studying in the rare disease, SCA type3.

The figures and the statistical methods are good.

The results and discussion have been presented well and findings summarized well in the conclusions.

Please consider this minor revision:

-The author described FD analysis as a quantitative indicator of object complexity which can be used to investigate morphological changes in the cerebral cortex and the one 3D-FD value for each parcellated region of the brain was obtained.

-The result (table 2) showed significantly atrophied brain regions in the patients with SCA type 3. >> To give the study face validity, please clearly describe or provide literature evidence that how the 3D-FD values represent degree of atrophy of the cerebral cortex.

Authors’ response:

We appreciate reviewer’s valuable suggestion. To demonstrate how FD can detect morphological change and the FD value can represent degree of atrophy of the cerebral cortex, please see the illustrated figure below which is from Fig.7 of our previous study published in NeuroImage, 2012, ” Fractal dimension analysis for quantifying cerebellar morphological change of multiple system atrophy of the cerebellar type (MSA-C)”. (please refer to author coverletter-394132, we have uploaded) Fig. (a) illustrates the FD of gray matter (GM) of cerebellum for MSA-C and Normal subjects. In Figure (a), the FD values which are below 2.5538 (2.4507, 2.4629, 2.4883 and 2.5392) present the FD values of GM for MSA-C patients, and others are for the normal subjects. Clearly, the MSA-C patients revealed atrophied GM of cerebellum than the normal subjects, and a smaller FD value indicates more atrophied GM in the cerebellum. Fig. (b) illustrates the FD of white  matter (WM) of cerebellum for MSA-C and Normal subjects. In Figure (b), the FD values which are below 2.2197 (1.9813, 2.0376, 2.1021 and 2.1624) present the FD values of WM for MSA-C patients, and others are for the normal subjects. Again, the MSA-C patients revealed atrophied WM of cerebellum than the normal subjects, and the more of decreased FD values indicate the more atrophied GM in the cerebellum.

In the revised manuscript, we have added a paragraph in the material and method section to scribe and provide literature evidence that how the 3D-FD values represent degree of atrophy of the cerebral cortex. The added paragraph is as follow:

“Our previous studies have demonstrated that the degree of cortical degeneration, which was quantified using FD analysis, can indicate the severity of neurodegenerative disorders [16]. Because FD analysis is based on a logarithmic scale, small increases in the FD value correspond to large increase in complexity. A higher FD value indicates a more complex cerebral structure, while a decrease in the cerebral FD value may indicate a degeneration of the cerebral structure [16]. Hence, this method can be used for effectively measuring structural cortical changes in both the cerebellar and extracerebellar regions and facilitated evaluating the severity of the cortical atrophy in this SCA3 study.”

Round  2

Reviewer 1 Report

Authors have now fixed the required minor editing and the paper runs better now